# Epigenetic Biomarkers of Metabolic Responses to Lifestyle Interventions

**DOI:** 10.3390/nu15194251

**Published:** 2023-10-03

**Authors:** Omar Ramos-Lopez

**Affiliations:** Medicine and Psychology School, Autonomous University of Baja California, Tijuana 22390, Mexico; oscar.omar.ramos.lopez@uabc.edu.mx; Tel.: +52-322-229-88-60

**Keywords:** epigenetics, biomarkers, metabolic alterations, DNA methylation, histone modifications, miRNAs, precision nutrition

## Abstract

Studies have examined the possible utility of epigenetic phenomena (DNA methylation changes, covalent histone modifications, and miRNA expression patterns) in predicting individual responses to different lifestyle programs. Nonetheless, most available evidence is focused on identifying epigenetic marks eventually associated with body composition and adiposity outcomes, whereas their roles in metabolic endings remain less explored. This document comprehensively reviewed the evidence regarding the use of epigenetic signatures as putative biomarkers of metabolic outcomes (glycemic, lipid, blood pressure, and inflammatory/oxidative stress features) in response to different lifestyle interventions in humans. Although more investigation is still necessary in order to translate this knowledge in clinical practice, these scientific insights are contributing to the design of advanced strategies for the precise management of cardiometabolic risk, gaining understanding on metabolic heterogeneity, allowing for the prediction of metabolic outcomes, and facilitating the design of epigenome-based nutritional strategies for a more customized approach for metabolic alterations treatment under the scope of precision nutrition.

## 1. Introduction

Epigenetics involves the study of heritable changes in gene expression patterns, which are not related to variations in the DNA sequence [1]. The main epigenetic signatures analyzed in medical sciences comprise DNA methylation (DNAm), histone modifications (acetylation/methylation), and long non-coding RNAs such as microRNA (miRNA) expressions [2]. Cumulative scientific evidence has elucidated the involvement of aberrant epigenetic phenomena in the development of metabolic disorders, including glucose intolerance, insulin resistance, dyslipidemia, high blood pressure, hyperinflammation, and oxidative stress [3]. Indeed, some of these epigenetic marks have been proposed as putative biomarkers of disease development [4].

Lifestyle advice (encompassing energy-restricted diets varying in macronutrient composition and the practice of regular physical exercise) continues to be the primary cornerstone in the prevention and non-pharmacological treatment of obesity and related metabolic alterations [5]. However, the beneficial effects of these therapies are heterogeneous, with a range of responses among individuals, suggesting the implications of additional extrinsic and intrinsic factors accounting for this variability [6].

Of note, clinical trials have examined the utility of epigenetic signatures in predicting individual responses to different lifestyle programs; most evidence is focused on identifying those epigenetic events associated with body composition and adiposity outcomes [7,8], whereas their roles in metabolic endings remain less explored.

This document comprehensively reviews the evidence regarding the use of epigenetic landscapes as putative biomarkers of metabolic features (with a focus on cardiometabolic risk factors) in response to different lifestyle interventions in humans, emphasizing their potential application as innovative tools for gaining understanding on metabolic heterogeneity, the prediction of metabolic outcomes, and the design of epigenome-targeted nutritional strategies for a more customized management of metabolic alterations under the scope of precision nutrition (Figure 1).

### 1.1. Epigenetic Biomarkers of Glycemic Outcomes

Glycemic control is a crucial goal in preventing the incidence of type 2 diabetes mellitus as well as delaying the occurrence of long-term complications. Specific epigenetic influences of glucose homeostasis are observed in several tissues and pathways associated with metabolism. Indeed, epigenetic mechanisms have been involved in the regulation of glucose metabolism insulin signaling, and the risk of type 2 diabetes mellitus. In this context, some trials have evaluated the utility of epigenetic marks as predictive tools for glycemic control after nutritional advice.

Findings from the POUNDS lost trial revealed significant interplays between dietary protein intake and DNAm levels on changes in insulin and HOMA-IR at 6 months of nutritional intervention, where mean protein intake (15%E) was associated with a greater decrease in insulin and HOMA-IR in participants with overweight or obesity within the highest tertile of DNAm at the *TXNIP* gene, but not in those with a high protein (25%E) intake [9]. Over a 12-month period of lifestyle modification (energy-restricted diet plus increased physical activity), changes in blood glucose levels were positively associated with the methylation levels of the *LINE-1* and *LUMA* (also known as *TMEM43* or transmembrane protein 43) genes in female breast cancer survivors with overweight and sedentary from different ancestries [10].

Meanwhile, participants with abdominal obesity who had more decreases in serum miR-99b had a lower fasting glucose, and higher HOMA of β-cell function after 18 months of an intervention trial comprising a hypocaloric diet plus regular physical activity [11]. In this same trial, circulating levels of the miR-128-1-5p interacted with physical activity on changes in HOMA-IR, finding that the miR-128-1-5p level increases after 6 months of consuming hypocaloric diets while different macronutrient contents were associated with smaller reductions in HOMA-IR in participants with no changes/decreases in their physical activity patterns [11]. Also, plasma miR-374a-5p negatively correlated with fasting insulin and HOMA-IR changes in obese/prediabetic patients from Asian/Indian ancestries following a lifestyle intervention (healthy diet plus daily physical activity) lasting 4 months [12]. Moreover, circulating miR-221-3p expression inversely correlated with a quantitative insulin sensitivity check index in Spanish girls with abdominal obesity who underwent 8 weeks of a multidisciplinary program comprising lifestyle education (based on a Mediterranean-style diet) and engaging in physical activity on a regular healthy basis [13]. In the same way, low blood miR-21/ROS/HNE levels were strongly associated with a reduction in the glycemic damaging axis in dysglycemic subjects after 1 year of habit intervention (Mediterranean diet), especially in those with values of 2h plasma glucose lower than 200 mg/dL [14].

Using a general linear model, it was demonstrated that a set of 12 circulating miRs (miR-106b, let-7c, miR-17, miR-296, miR-20b, miR-92a, miR-93, miR-186, miR-192, miR-342, miR-374b, miR-22) could predict fasting blood glucose changes in adults at risk of suffering type 2 diabetes mellitus who underwent a 12-week behavioral intervention including nutrition and physical activity advice [15]. Patients with coronary heart disease displaying low plasma levels of miR-145 at baseline showed a higher risk of developing type 2 diabetes after consumption of a low-fat (<30%E)/high complex carbohydrate (55%E) diet over 60 months [16]. Meanwhile, within this same sample, patients with low blood concentrations of miR-126, miR-29a, and miR-28-3p also presenting high blood levels of miR-150 at baseline had an increased risk of type 2 diabetes after consuming a Mediterranean diet [16]. Furthermore, miR-34a and miR-210 variations positively correlated with glucose levels changes, whereas miR-34a negatively correlated with changes in insulin levels in overweight/obese subjects who followed a calorie-restricted diet with a different macronutrient distribution for 6 months [17].

Overall, these findings highlight the utility of gene-specific methylation patterns, global methylation, and miRNA expression patterns as biomarkers of glycemic trajectories following dietary counseling (mainly Mediterranean diets). Further studies evaluating different nutritional prescriptions considering geographic/regional differences are required.

### 1.2. Epigenetic Biomarkers of Lipid and Lipoprotein Outcomes

Dyslipidemia (imbalance of blood lipids) is a widely known risk factor for cardiovascular disease. Several factors, including epigenetic phenomena and lifestyle habits, influence lipid homeostasis. Thus, some nutriepigenetic studies have analyzed the impact of epigenome–nutrient interactions on blood lipid outcomes in some populations.

In this context, overweight/obese participants who followed a low-fat diet (20%E) and concomitantly presented a higher regional DNAm level at the *CPT1A* gene underwent a greater reduction in total plasma triglycerides and related lipoproteins compared with those who consumed a high-fat diet (40%E) after 2 years of intervention [18]. Baseline methylation in the *BMAL1* gene negatively correlated with the effect of a 4-month intervention based on an energy-controlled Mediterranean dietary pattern on serum lipids, including the percentage of change in total cholesterol and LDL-c in overweight/obese women [19]. The methylation status of the insulin gene was related to changes in LDL-c after an intervention with inulin agave (10 g/d) over 2 months in patients with type 2 diabetes [20]. Concomitant reductions in the methylation levels of the *ADRB3* gene and increases in the HDL-c values were found in overweight/obese women who consumed a diet fortified with folate (191 μg/d) and hazelnut oil (1 capsule/d) for 8 weeks [21]. Moreover, *LINE-1* methylation levels at baseline were associated with variations in the blood triglyceride levels in volunteers with metabolic syndrome who followed an energy-restricted dietary pattern for 8 weeks [22].

Reductions in the intrahepatic/pancreatic fat depots after 18 months of a lifestyle intervention (hypocaloric diet plus physical activity) were related to greater circulating decreases in miR-100 or miR-99a among adults with abdominal obesity [23]. Plasma decreases in miR-100 were also related to reductions in ectopic fat depots in other organs such as the kidney, muscle, and heart at the end of the intervention [23]. A negative correlation between adipose-miR-329-3p expression and HDL-c concentrations was found in women who underwent supervised aerobic/resistance training over 12 weeks [24]. Decreases in the plasma concentrations of miR-330-3p and miR-125a-5p mirrored the benefits in the lipid profile in healthy women who followed an 8-week trial with a normocaloric diet enriched with PUFAs (30 g/day of almonds and walnuts), being inversely associated with total cholesterol, triglycerides, and VLDL-c [25]. Changes in serum miR-130b and miR192 were associated with decreases in VLDL-c, the latter also being associated with changes in fasting triglycerides in this same study [25]. The circulating miR-185-5p difference in expression was related to the HDL-c variation after 6 months of Mediterranean dietary intervention in BRCA-mutated women [26]. Variations in the HDL-miR-223-3p plasma levels negatively correlated with differences in serum HDL-c in men after consuming an isoenergetic diet high in industrial *trans* fatty acids (3.7%E) for 4 weeks [27]. In this same trial, changes in the HDL-miR-135a-3p concentrations positively correlated with fluctuations in the total blood triglycerides following the intake of industrial *trans* fatty acids, and with variations in the low-density lipoprotein–triglyceride concentrations following the high consumption of *trans* fatty acids (3.7%E) from ruminant sources [27].

In summary, the methylation status of genes modulating fatty acid transport and temporal orchestration of fat metabolism as well as circulating miRNA levels have been associated with inter-individual differences in blood triglycerides and lipoprotein concentrations after energy-restricted diets varying in fat content. Further validation of these results is needed in order to use these epigenetic signatures as biomarkers of lipid control in the clinical setting.

### 1.3. Epigenetic Biomarkers of Blood Pressure Outcomes

Increasing evidence supports that epigenetic modifications are implicated in hypertension pathogenesis. Therefore, research in this field is providing insights into the importance of blood pressure regulation via different mechanisms. In this regard, epigenetic marks as surrogates of blood pressure changes after nutritional interventions have been evaluated.

For example, a higher regional DNAm at the *LINC00319* gene was associated with greater reductions in systolic and diastolic blood pressures in overweight/obese adults responding to a 2-year low-fat diet (20%E) [28]. More importantly, participants with a higher DNAm at *LINC00319* were more likely to maintain these blood pressure changes after the dietary intervention [28]. Genome-wide methylation analysis in mononuclear cells from infants identified some DNAm sites mapped to metabolic genes (*ACOT13*, *KCNV1*, and *NR3C1*), nucleic acid binding proteins (*RBM9* and *CDYL*), and transcription factors (*GATA4* and *TBPL2*) putatively associated with phenotypic changes mainly in blood pressure after 9 months of fish oil (3.8 g/d) supplementation [29]. However, because the postnatal effects of fatty acids on DNAm profiles were small, additional research is needed to verify these results. Interestingly, it was reported that performing sports activities (i.e., aerobic and resistance training) at early ages may decrease the risk of suffering hypertension in later life, probably involving DNAm changes in genes influencing several pathways related to cardiovascular function such as aldosterone-regulated sodium reabsorption, inflammatory mediator regulation of TRP channels, vascular smooth muscle contraction, cholinergic synapse, among others [30].

After 4 months of lifestyle intervention (improvement in diet quality plus increase in daily physical activity), circulating miR-1-3p variations negatively correlated with systolic blood pressure changes in Asian/Indian patients with obesity and prediabetes [12]. Indeed, it was postulated that a possible mechanism underlying the beneficial effect of 6-week supplementation with the mitochondria-targeted antioxidant “MitoQuinol” (20 mg/d) and moderate endurance training on reduced blood pressure in hypertensive individuals could be via an increase in the expression of miR-126 [31].

Although there is some evidence of epigenetic events associated with nutrition-related blood pressure outcomes, most of the available studies have found them as chance finds or secondary results. Therefore, studies aimed at identifying specific epigenetic changes in relevant genes related to hypertension or blood pressure homeostasis are warranted.

### 1.4. Epigenetic Biomarkers of Inflammatory and Oxidative Stress Outcomes

Chronic and persistent inflammatory processes may induce oxidative stress states via the overproduction of free radicals and reducing the cellular antioxidant capacity, which are involved in the development of many diseases, including obesity, diabetes, and cardiovascular diseases. Interestingly, epigenetic mechanisms have been implicated in the development and function of the immune system as well as in the regulation of free radicals production via the mitochondrial metabolism. Thus, some nutritional programs have targeted inflammatory/stress events for preventive and treatment strategies, where epigenetic signatures may play a role.

For example, changes in leukocyte *TNF* and *IL6* DNAm following 8 weeks of eccentric resistance-training exercise negatively correlated with TNF expression in healthy, non-smoking males with no history of resistance-training exercise [32]. Altered cytokine secretion in response to a diabetes support program (based on standard diabetes self-management education guidelines focused on reducing risk factors associated with diabetes complications) for 3 months was significantly related to modifications in the methylation/expression status of immune-related genes (i.e., *FCER2*, *DUSP10*, *BCL2*, and *CD6*) in monocytes from Native Hawaiian adults with diabetes mellitus [33]. Also, correlations between CRP protein levels and methylations of *IL-6*, *CRP,* and *LINE-1* genes were detected after an 8-week yoga intervention in a community population of women reporting psychological distress [34]. Epigenome-wide association analysis identified several CpG sites (including the site cg07030336 annotated to the *VTI1A/ZDHHC6* gene) whose methylation levels were associated with accelerometer-assessed total physical activity and inflammatory markers such as CRP in a general Japanese population [35]. Overweight/obese women supplemented with folate and a hazelnut oil capsule (average 191 μg/d of folate and 1 hazelnut oil capsule) for 8 weeks resulted in a decrease in the blood concentrations of malondialdehyde and an increase in the total antioxidant capacity (both being used to evaluate oxidative stress) together with a decrease in *ADRB3* DNAm levels [21]. Continued high-intensity interval walking and moderate exercise appeared to attenuate the age-dependent reduction in *ASC* methylation, suppressing the excess pro-inflammatory cytokines such as IL-1beta and IL-18 [36]. Likewise, findings from the Cardiovascular Health Study reported that performing higher physical activity was associated with an anti-inflammatory state by inducing *IL-10* hypomethylation and *TNF* hypermethylation [37].

Between 4 and 8 months of follow-up to a lifestyle program (healthy dietary recommendations plus exercise training), the serum variations in miR-128-3p negatively correlated with changes in the blood concentrations of MCP-1 (an inflammatory marker) in Asian Indian adults with obesity and prediabetes [12]. The role of circulating miRNAs as early indicators of a 12-week diet (plant-based eating) and physical activity (strength training) responses in women with metastatic breast cancer was explored, where the expression levels of miR-10a-5p and miR-211-5p were downregulated in women who displayed a poor response to the intervention, evidenced by no changes or increased levels of the pro-inflammatory cytokines IL-6, TNF-α, and CRP [38]. Instead, a successful inflammatory response (characterized by decreased levels of IL-6, TNF-α, and CRP) was related to the upregulation of these biomarkers [38]. Dietary modification (8-week normocaloric diet plus 30 g/day of almonds and walnuts) of plasma miR-130b and miR-221 concentrations positively mirrored changes in CRP to some extent in healthy women [25]. Increasing the serum miR-15 and miR-21 values showed a significant inverse association with IL-6 expression in lymphocytes from subjects with prediabetes who consumed a pistachio-supplemented diet (57 g/d of pistachios) for 4 months [39]. Subjects with lower miR-126 showed a significant positive association with IL-6 in these same cells [39]. Variations in plasma miR-223-3p concentrations positively correlated with variations in CRP concentrations after a high intake (3.7%E) of ruminant *trans* fatty acids for 4 months in men [27]. Within a 6-week double-blind, randomized clinical trial, supplementation with the mitochondria-targeted antioxidant “MitoQuinol” (20 mg/d) and moderate endurance training, individually and more prominently in combination, could reduce IL-6 and concomitantly increase the antioxidant capacity serum levels in individuals with hypertension, in association with lowering serum miR-21 and miR-222 levels [40]. Whereas miR-222-3p negatively correlated with the plasma levels of TNF-α and IL-6, and miR-103a-3p expression negatively correlated with the serum TNF-α concentrations in the pregnancy stage and 2–3 years post-delivery period in women with gestational diabetes mellitus who consumed a Mediterranean diet supplemented with extra virgin olive oil (>40 mL/d) and pistachios (>35 g/d) [41]. Furthermore, a ketogenic diet (<30 g of carbohydrates/d) could normalize the expression of miRNAs linked to antioxidant/anti-inflammatory pathways in subjects with obesity when compared with normal-weight subjects, where the upregulation of hsa-miR-30a-5p correlated with the reduction in catalase protein expression in red blood cells [42].

In conclusion, some studies support the role of epigenetic modifications in modulating inflammatory and oxidative stress responses to nutritional treatments, which are located in important genes directly related to immune function and inflammation. The evaluation of the effect of supplementation with vitamins, minerals, and bioactive compounds with antioxidant and anti-inflammatory function is warranted.

### 1.5. Epigenetic Biomarkers of Adipokine Outcomes

Adipokine secretions derived from altered adipose tissues play a pivotal role in the development of metabolic disorders. Indeed, the aberrant production/release of adipokine represent a possible link between obesity and the risk of developing dyslipidemia, inflammation, and insulin resistance. Remarkably, adipokine expression may be linked to epigenetic regulation and nutritional influences. Consistently, some trials have explored the clinical value of epigenetic phenomena to monitor the improvements in adipokine concentrations under certain dietary conditions.

Of note, several methylation sites annotated to a genomic region on chromosome 6 encompassing the *RNF39* gene negatively correlated with adiponectin blood levels post an 18-month intervention with a Mediterranean dietary pattern plus low-carbohydrate or low-fat regime with or without physical activity in overweight individuals [43]. Also, the methylation rates of the *COL4A1* gene negatively correlated with a serum leptin change within the DIRECT PLUS trial comprising a Mediterranean diet enriched with 1240 mg of polyphenols [44]. Moreover, a weight-loss nutrition intervention supplemented with fish oil (6 capsules/d) for 8 weeks could reduce circulating leptin in overweight women, with adjacent modifications in the DNAm of *CD36*, *CD14*, *FADS1*, and *PDK4* genes in white blood cells [45].

Interestingly, miR-587 expression positively correlated with the serum leptin levels in patients with metabolic syndrome features consuming the RESMENA diet (consisting of high meal frequency and enhanced adherence to the Mediterranean diet), suggesting a mediation role of this miRNA in metabolic adaptations following dietary modifications [46]. Similarly, relevant correlations between miR-103a-3p relative changes and blood leptin and orexin outcomes were found in children with obesity trained in a program of exercise and dietary counseling for 6 weeks [47]. Accordingly, miR_22_3p positively correlated, whereas miR_17_5p negatively correlated with blood leptin levels after a controlled weight-loss trial (exercise plus nutrition advice) for 6 months in female breast cancer survivors [48]. In addition, plasma fluctuations of miR-128-3p negatively correlated with adiponectin changes in obese/prediabetic individuals submitted to dietary and physical activity modifications for 4 months [12].

Together, some results suggest associations between epigenetic status and changes in the levels of adipokines after nutritional therapies. However, most of the identified genes are not directly related to adipokine production/regulation, pointing to indirect effects and alternative mechanisms. Future research addressing these aspects is necessary.

## 2. Future Directions

The integration of multiple omics knowledge has emerged as a ground-breaking approach to providing a more holistic view of the molecular mechanisms underlying human metabolic disorders as well as for improving the prediction of disease risk and therapy responsiveness between individuals for precision nutrition [49]. In this context, epigenome phenomena play a key role in modulating cell phenotypes through regulating the expression of relevant genes implicated in physiological and pathological processes, where nutritional features behave as potential intermediate orchestrators [50]. In turn, the metabolism of dietary compounds (i.e., fiber, fats, and polyphenols) has an effect on gut microbiome composition and function, including the production of specific metabolites such as short-chain fatty acids and bile acids, which can modify host cell responses to environmental stimuli via epigenome modifications [51]. Reciprocally, epigenetic processes may influence microbiota status, affecting the transcription of certain microbial genes that control the growth bacterial species and shaping the structure of the intestinal microbial community [52]. Moreover, a number of dietary metabolites can modulate the activity of epigenetic enzymes (i.e., DNA methyltransferases and histone methyltransferases/deacetylases) and related transcriptional functions to maintain cell homeostasis, synchronizing the expression state of chromatin with metabolism rates [53]. Furthermore, some genome sequence variants have been associated with particular DNA methylation changes, histone marks, or miRNA expression patterns, which can induce rearrangements of chromatin architecture, with long-range transcriptional effects and health impacts [54].

Based on these findings, studies combining epigenome insights with other complementary biological information have been performed. For instance, interactions between circulating miRNAs (miR-130b-3p, miR-185-5p, and miR-21-5p) and the abundance of gut bacterial species (i.e., *Bacteroides eggerthi*) in relation to obesity development were found, suggesting an epigenetic bridge between the gut microbiome and human host in adiposity status [55]. Also, complex relationships between dietary folate intake, *CAMKK2* DNAm and expression levels, and insulin resistance were reported [56]. Meanwhile, it was demonstrated that acute aerobic exercise followed by the supplementation of n-3 polyunsaturated fatty acids (5.7 g/d) and extra virgin olive oil (6 g/d) for 4 weeks induced DNAm changes in leukocytes associated with inflammatory and oxidative stress markers via the modulation of DNMT mRNA expression in leukocytes from trained male cyclists [57]. Similarly, the success of a weight-loss program (hypocaloric diet with moderately high-protein content) on adiposity measurements was related to differential DNAm and expression levels of the *CD44* gene on white blood cells [58]. Of note, methylome and transcriptome changes in the *ZNF331* and *FGFRL1* genes in leukocytes underlined the effects of a very-low-calorie ketogenic diet (600–800 kcal/d) on obesity indices [59]. Moreover, endurance exercise positively induced DNAm with a corresponding decrease in mRNA expression in human adipose tissue involving genes such as *RALBP1*, *TCF7L2*, *KCNQ1*, *GABBR1*, *EHMT1*, *EHMT2*, *HDAC4,* and *NCOR2*, providing an integrated molecular mechanism modulating adipocyte metabolism after exercise [60]. Additionally, supplementation with riboflavin (1.6 mg/d for 16 weeks) altered the DNAm status of hypertension-related genes (i.e., *IGF2*, *ACE*, *GNA12*, and *AGTR1*) in carriers of the *MTHFR* 677TT genotype, revealing an epigenomic mechanism linking hypertension risk with the allele-specific response of blood pressure to riboflavin intake [61]. Although more investigations are needed, these results highlight the importance of linking diverse types of biological data (including epigenomics) to expanding the current understanding of human metabolism complexity and diversity, yielding profound insights into disease pathogenesis and precise management.

## 3. Conclusions

There is a high variability in metabolic responses to lifestyle advice, which supports the need to search for factors accounting for this heterogeneity that could be useful for the personalization of treatments according to particular phenotypes. To date, there is evidence of associations between epigenetic signatures and metabolic outcomes after lifestyle interventions. These include DNAm levels in genes related to glycemic (*TXNIP*, *LINE-1*, and *LUMA*), lipid (*CPT1A*, *BMAL1*, *ADRB3*, and *LINE-1*), blood pressure (*LINC00319*, *ACOT13*, *KCNV1*, *NR3C1*, *RBM9*, *CDYL*, *GATA4*, and *TBPL2*), inflammatory/oxidative stress (*TNF*, *IL6*, *FCER2*, *DUSP10*, *BCL2*, *CD6*, *CRP*, *LINE-1*, *VTI1A/ZDHHC6*, *ADRB3*, and *ASC*), and adipose tissue-related adipokine (*RNF39*, *COL4A1*, *CD36*, *CD14*, *FADS1*, and *PDK4*) changes after the consumption of specific hypoenergetic diets (occasionally supplemented with functional foods/bioactive compounds) or systematic physical exercises. Moreover, the expression of particular miRNAs correlated with the response rates in glycaemia (miR-99b, miR-128-1-5p, miR-374a-5p, miR-221-3p, miR-21/ROS/HNE, miR-106b, let-7c, miR-17, miR-296, miR-20b, miR-92a, miR-93, miR-186, miR-192, miR-342, miR-374b, miR-22, miR-145, miR-126, miR-29a, and miR-28-3p, miR-150, miR-34a, and miR-210), lipid profile (miR-100, miR-99a, miR-329-3p, miR-330-3p and miR-125a-5p, miR-130b, miR192, miR-185-5p, miR-223-3p, and miR-135a-3p), blood pressure (miR-1-3p and miR-126), inflammation (miR-128-3p, miR-10a-5p, miR-211-5p, miR-221, miR-15, miR-21, miR-223-3p, miR-21, miR-222, miR-222-3p, miR-103a-3p, and hsa-miR-30a-5p), and adipokine secretion (miR-587, miR-103a-3p, miR_22_3p, miR_17_5p, and miR-128-3p) in some populations.

Overall, this knowledge is helping to understand the heterogeneous responses to lifestyle treatments, where epigenetic phenomena may play an important modulatory role. Moreover, available studies suggest that such epigenetic marks can serve as potential biomarkers of therapy success in populations with compatible characteristics. However, more research is still necessary to translate this information into real clinical practice, considering aspects of variability such as the type of population for validation, the coexistence of metabolic risk factors, lifestyle intervention characteristics (type of treatment, dosage, time of follow-up), and origin of the analyzed samples. Also, investigation regarding the role of histone modifications is warranted. Additionally, interactions of the epigenome with other molecular areas, including genomics, metagenomics, transcriptomics, proteomics, and metabolomics, need special attention. In any case, these scientific insights are contributing to the design of innovative predictive strategies for the precise management of cardiometabolic risk.

## Figures and Tables

**Figure 1 nutrients-15-04251-f001:**
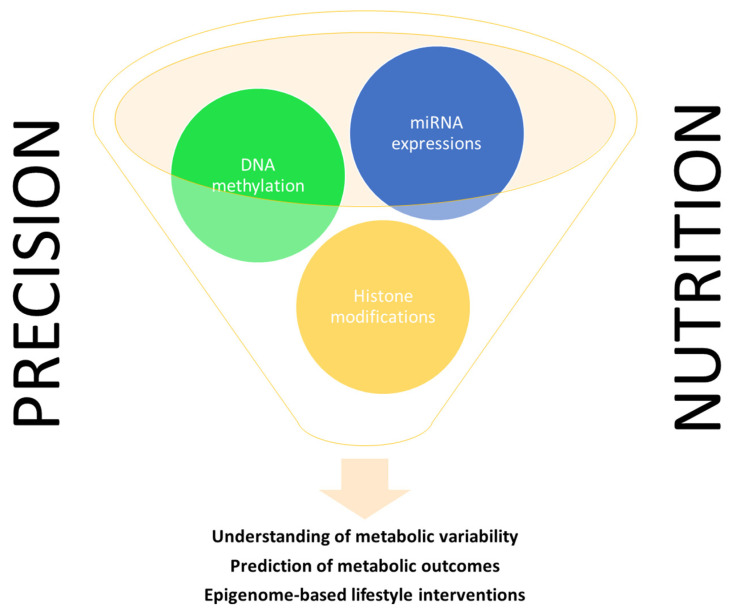
Epigenetic signatures as biomarkers of metabolic responses to lifestyle interventions.

## Data Availability

Not applicable.

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
