# Peer review of "Epigenetic Biomarkers of Metabolic Responses to Lifestyle Interventions"

_nutrients, 2023, doi:10.3390/nu15194251_

Round 1
Reviewer 1 Report
This has potential as a review however the sections are an uncoordinated collection of statements from papers and studies. There does not appear to be a cohesive story or message from each section that allows the reader to determine the biomarkers that would be of interest within populations or disease states. For example there is a reference to infants with hypertension(?) given 9 months of fish oil supplementation where DNA methylation targets were identified. This study showed a comparison between fish oil and sunflower oil that showed very small differences that may not be impactful without further study. Further none of the genes described in that study are defined in this section. Then the author jumps to the benefit of exercise at a young age to BP control. The paragraphs throughout are a collection of one sentence summaries that do not provide a proper review of the subject nor a cohesive and clear argument in favor of biomarker monitoring as the title implies
Each section is a collection of perhaps interesting findings but the author has not put the narrative together in a way which argues in favor of epigenetic biomarkers as a means for monitoring the success of lifestyle interventions in metabolic disease. As a review, significant modification and restructuring should be considered
The conclusion should have discussed the genes and biomarker targets that would be useful in monitoring of chronic disease. I was left with no clear summary of the important biomarkers for metabolic monitoring during lifestyle intervention.
Many words are used incorrectly and are confusing- for example
"Adult women with overweight or obesity"
Author Response
Thank you for your comments!
Please, find attached the author responses.

Reviewer 2 Report
This article discusses the role of epigenetic markers in various health outcomes, including glycemic control, lipid and lipoprotein outcomes, blood pressure regulation, inflammatory and oxidative stress outcomes, and adipokine levels. Epigenetic markers are shown to be associated with these health outcomes in response to dietary and lifestyle interventions. For example, specific epigenetic markers are linked to changes in insulin and HOMA-IR in individuals with overweight or obese after nutritional intervention. Additionally, epigenetic markers are related to changes in blood glucose levels, lipid profiles, blood pressure, and inflammatory markers in response to various dietary and lifestyle modifications. Furthermore, the expression of certain microRNAs (miRNAs) is correlated with adipokine levels, such as leptin and adiponectin, in individuals undergoing dietary and exercise interventions. These findings highlight the potential of epigenetic markers and miRNAs as predictive tools for assessing and improving various health outcomes in individuals with different health conditions.The logic of this paper is clear, but there are still some issues that need to be addressed, such as:
1. The manuscript focuses on listing the research results of others, but lacks a summary and induction of these research results.
2. The relationship between different lifestyles and changes in 5 types of biomarkers was not clearly discussed。
3. Why choose these 5 types of results as hypothetical biomarkers, based on what?
4. Lacking the authors' own insights on the manuscript topic, what direction do they believe future research in this area should focus on?
5. Is this manuscript distinguishable from other recent reviews on this topic?
Author Response

(The authors gave the same response as above.)

Round 2
Reviewer 1 Report
much improved text and presentation. thank you for considering my comments and suggestions
Reviewer 2 Report
No more comments